# Proportional Stroke Mortality in Espírito Santo, Brazil: A 20-Year Joinpoint Regression Study

**DOI:** 10.3390/epidemiologia6020028

**Published:** 2025-06-19

**Authors:** Casanova André Motopa Mpuhua, Orivaldo Florencio de Souza, Blanca Elena Guerrero Daboin, Italla Maria Pinheiro Bezerra, Marcelino Na Blei, Thiago Dias Sarti, Vithor Ely Bortolin da Silva, Luiz Carlos de Abreu

**Affiliations:** 1Centro de Ciências da Saúde, Universidade Federal do Espírito Santos (UFES), Vitoria 29075-910, ES, Brazil; mpuhuacasanovaa@gmail.com (C.A.M.M.); bgdaboin@yahoo.com (B.E.G.D.); blei.marcelino121@gmail.com (M.N.B.); thiago.sarti@ufes.br (T.D.S.); 2Centro de Ciências da Saúde e do Desporto, Universidade Federal do Acre (UFAC), Rio Branco 69920-900, AC, Brazil; orivaldo.souza@ufac.br; 3Laboratório Multidisciplinar de Estudos e Escrita Científica em Ciências da Saúde (LaMEECCS), Universidade Federal do Acre (UFAC), Rio Branco 69920-900, AC, Brazil; 4Programa de Pós-Graduação em Políticas Públicas e Desenvolvimento Local, Escola Superior de Ciências da Santa Casa de Misericórdia de Vitória, Vitória 29045-402, ES, Brazil; italla.bezerra@emescam.br; 5Programa de Pós-Graduação em Ciências Médicas, Faculdade de Medicina da Universidade de São Paulo (FMUSP), São Paulo 05508-220, SP, Brazil; vithorely@gmail.com

**Keywords:** stroke, mortality, proportional mortality, epidemiology, trends, joinpoint regression

## Abstract

**Introduction:** Stroke is one of the leading causes of death and disability worldwide. In Brazil, it remains the primary cause of mortality among adults. Although overall stroke mortality rates have declined, the absolute number of stroke incidents, deaths, and years of life loss continues to rise, particularly in developing and underdeveloped countries. **Objective:** The aim of this study was to analyze trends in stroke mortality across different age groups and both sexes in Espírito Santo, Brazil, from 2000 to 2021. **Methods:** This ecological time series study utilized secondary data from Espírito Santo, Brazil, from 2000 to 2021. Mortality data, categorized by sex and age group, were obtained from the Department of Informatics of the Unified Health System (DATASUS) database. Stroke-related mortality included deaths recorded under the International Classification of Diseases, 10th Revision (ICD-10) codes for subarachnoid hemorrhage (I60), intracerebral hemorrhage (I61), cerebral infarction (I63), and stroke not specified as hemorrhagic or ischemic (I64). Temporal trends in stroke mortality were assessed using joinpoint regression analysis. **Results:** From 2000 to 2021, there was a significant reduction in proportional mortality from stroke, with an overall decrease of −3.7% (*p* < 0.001). When analyzed by sex, the decline was −3.0% (*p* < 0.001) for males and −3.9% (*p* < 0.001) for females. The most significant decrease in proportional mortality was observed in the 50 to 59 age group, with an average annual percentage change of −4.9% (*p* < 0.001). The 30 to 39 age group exhibited the smallest decline, with an average annual percentage change of −2.4% (*p* < 0.001). No significant segments were observed in the 40 to 49, 60 to 69, and 70 to 79 age groups during the study period. **Conclusions:** This study identified a notable decline in stroke-related proportional mortality in the adult population of Espírito Santo between 2000 and 2021. While males had a higher absolute number of deaths, females exhibited a higher proportional mortality rate, underscoring the need for targeted preventive measures and effective acute stroke treatment, particularly among men.

## 1. Introduction

Stroke is a significant public health concern. It is a severe condition that can cause permanent brain damage or death if not treated promptly. Stroke is characterized by a series of cerebrovascular events resulting from a disruption in cerebral blood flow and is associated with high rates of morbidity and mortality [1,2]. Despite advances in acute stroke treatment, mortality rates remain high, suggesting that current primary prevention strategies for stroke and cardiovascular diseases are either not widely implemented or insufficiently effective. In response, the World Health Organization (WHO) emphasizes the urgent need for preventive measures and effective treatments for this condition [3].

Globally, approximately 17 million stroke cases are reported each year. Of these, 6.5 million results in death, while the remainder contribute to the increasing prevalence of the disease [4]. In Brazil, stroke is the leading cause of death and disability among individuals over 50 years of age and accounts for nearly 40% of early retirements [5]. Moreover, the incidence of stroke is rising [6]. According to the WHO, this trend is expected to continue until at least 2060, by which time stroke will be responsible for 12.8% of all deaths in Brazil [7].

Espírito Santo, a state in Brazil’s southeast region, offers an important setting for studying stroke epidemiology in a mid-sized population context. Although it represents only about 2% of the country’s population [8], studying this state provides critical insights into stroke risk factors and control strategies. While the southeast region accounts for nearly 43% of Brazil’s population [9], most research focuses on larger metropolitan areas such as São Paulo, Rio de Janeiro, and Belo Horizonte. This makes research in smaller regions like Espírito Santo especially valuable for addressing knowledge gaps and reducing regional disparities. In this context, time-series regression models are effective tools for estimating population parameters, identifying trends, and supporting evidence-based public health decision-making [10]. Leveraging existing national and international evidence, it has been observed that proportional mortality due to stroke declines over time, with variations across sex and age groups. Therefore, this study set out to evaluate stroke mortality trends by age group and sex in Espírito Santo, Brazil, from 2000 to 2021.

## 2. Methods

### 2.1. Study Design and Population

This study is an ecological time-series analysis using secondary data on stroke-related deaths among adults residing in Espírito Santo, Brazil, from 2000 to 2021. In 2021, the adult population was approximately 2,973,566, with a Human Development Index (HDI) of 0.740 [11]. As this is a study based on data from the DATASUS public health database, the inclusion and exclusion criteria are limited to the variables available in the system.

The inclusion criteria consisted of all registered stroke-related deaths (ICD-10 codes I60–I64) among individuals aged 20 years and older, residing in the state of Espírito Santo, Brazil, between 2000 and 2021.

The exclusion criteria were records for individuals younger than 20 years or cases with missing geographic information that would prevent confirming residency in Espírito Santo.

### 2.2. Data Extraction

Data on stroke-related deaths and all-cause mortality were obtained from the Information Technology Department of the Unified Health System (DATASUS) of the Brazilian Ministry of Health [12]. The health professionals issue death certificates, which are subsequently entered into the Mortality Information System and transferred to the DATASUS database. Population count data were also extracted from the DATASUS database, available at https://datasus.saude.gov.br/populacao-residente (accessed on 22 November 2023). All data were retrieved using the Tabnet tabulator tool provided by the Brazilian Ministry of Health.

As the data are publicly available and do not contain any personal identifiers, individual patient consent and ethical approval were not required for this study. All analyses were conducted following the ethical guidelines and regulations governing the use of secondary data for research purposes.

### 2.3. Study Variable

The primary outcome variable of the study was mortality due to stroke. Stroke mortality data were presented for all adults and stratified by sex (male and female) and age groups (20–29 years, 30–39 years, 40–49 years, 50–59 years, 60–69 years, 70–79 years, and 80 years and older).

The criteria of the American Heart Association and the American Stroke Association were adopted to define stroke-related deaths [1]. According to these criteria, stroke is defined as a neurological deficit attributed to an acute focal injury of the central nervous system of vascular origin, encompassing cerebral infarction, intracerebral hemorrhage, and subarachnoid hemorrhage. Consequently, stroke-related mortality included deaths recorded under the International Classification of Diseases, 10th Revision (ICD-10) codes for subarachnoid hemorrhage (I60), intracerebral hemorrhage (I61), cerebral infarction (I63), and stroke not specified as hemorrhagic or ischemic (I64). This classification also follows the reporting standards of the Brazilian Ministry of Health and reflects how stroke mortality is aggregated in the national DATASUS.

Although the pathophysiological mechanisms of these stroke subtypes differ, it is methodologically consistent and accepted in ecological time-series studies to analyze them as a unified outcome [13,14]. This approach enables comparability with another population-level research using the same database. Therefore, stroke was analyzed as a single outcome variable in line with the study’s ecological design and population-level objectives.

Proportional mortality was calculated by dividing the number of stroke deaths by the total number of deaths from all causes and then multiplying the quotient by 100; hence, it is expressed as a percentage.

### 2.4. Statistical Analysis

The temporal trend of proportional mortality from stroke was assessed using the Average Annual Percentage Change (AAPC) and Annual Percentage Change (APC) metrics. This analysis was conducted with the Joinpoint Regression Program (version 5.0.2, 2023), developed by the National Cancer Institute, Rockville, MD, USA [15].

Joinpoint regression models were employed to identify significant changes in the time series data and to determine the trend within each segment of proportional mortality from 2000 to 2021 [15]. This method is commonly used in epidemiological studies to analyze trends in disease incidence and mortality [16,17]. It identifies specific points in time—called joinpoints—where a statistically significant change in the trend occurs [18]. The model fits a series of connected linear segments to the time series data, with each segment representing a distinct trend. This approach allows researchers to detect changes in the direction or rate of trends over time, such as periods of acceleration, deceleration, or reversal. The Joinpoint Regression Program selects the most appropriate model, using the Bayesian Information Criterion (BIC).

BIC helps choose the best model by favoring those that explain the data well without adding unnecessary joinpoints. Lower BIC values indicate better models. For each segment, 95% confidence intervals were calculated using the parametric method, and statistical significance was defined as a *p*-value ≤ 0.05.

### 2.5. Data Quality and Missing Data Handling

To address missing data, we applied complete case analysis, following recommendations [19], which consider this approach acceptable when the dataset exceeds 1000 observations and the proportion of missing data is under 20%. Given the minimal proportion of missing cases (0.018%), the application of this method is unlikely to have introduced bias or affected the validity of the results.

## 3. Results

Between 2000 and 2021, 26,907 stroke-related deaths were analyzed. The sex variable was missing in five records, corresponding to 0.018% of the dataset, specifically in the years 2001, 2004, 2006, 2018, and 2019. These omissions are considered negligible, with no documented evidence of systemic registration failures or clinical anomalies.

Of the total deaths, 13,884 (51.6%) occurred among males. Individuals aged 70 years and older accounted for 58.2% of all stroke-related deaths.

As illustrated in Table 1, the years with the highest number of stroke-related deaths were 2000 (1482 deaths), 2002 (1497 deaths), 2003 (1496 deaths), and 2019 (1422 deaths). Among males, the highest numbers were recorded in 2000 and 2002, with 808 and 816 deaths, respectively. For females, the peak years were 2003 (700 deaths) and 2019 (723 deaths). The lowest number of stroke-related deaths occurred in 2016, with 540 deaths among males and 504 among females.

Proportional mortality from stroke among adults declined significantly, from 9.47% in 2000 to 4.44% in 2021. As presented in Table 2, females consistently exhibited higher proportional mortality from stroke than males throughout the study period. The highest proportional mortality for both sexes was recorded in 2000, while the lowest levels were observed in 2021. The age groups 70–79 years and 80 years and older registered the highest proportional mortality rates, whereas the 20–29-year-old group consistently reported the lowest rates.

From 2000 to 2021, proportional mortality from stroke declined by an average of 3.7% per year. The reduction was more pronounced among females than males. The most significant annual decreases were observed in the 50–59 and 40–49 age groups, both averaging a 4.9% decline. In contrast, the 20–29 age group was the only one to display a stable trend over the study period. Detailed results are presented in Table 3.

Figure 1 illustrates the annual percentage change in proportional mortality from stroke among adults, stratified by sex. For the overall adult population, two joinpoints were identified, with statistically significant declines observed in the 2000–2016 and 2019–2021 segments (*p* ≤ 0.05). Among males, a significant annual decline of −4.0% was observed from 2000 to 2015, followed by a stable trend. In females, three joinpoints were identified, with significant declines of −3.6% (2000–2011), −6.9% (2011–2016), and −16.5% (2019–2021).

The temporal trend in proportional mortality from stroke, stratified by age group, demonstrated several significant patterns over the study period. Statistically significant declines were identified in the intervals 2000–2011, 2011–2016, and 2019–2021 (*p* < 0.05). The age groups 40–49, 60–69, and 70–79 did not show any joinpoints but exhibited steady and significant declines throughout 2000–2021. In contrast, the 20–29 age group also showed a single uninterrupted segment, though the trend was not statistically significant (*p* > 0.05). Among individuals aged 30–39, a significant annual decrease of 4.3% (*p* < 0.05) was followed by a period of stability. The 50–59 age group presented two joinpoints, but only the 2000–2016 segment showed a statistically significant decline. For those aged 80 and older, three joinpoints were observed, with significant reductions of 2.8% per year from 2000 to 2010 and 7.1% per year from 2010 to 2016 (*p* < 0.05). These trends are visually presented in Figure 2.

## 4. Discussion

Our study’s primary findings reveal a consistent decline in the temporal trend of proportional stroke mortality across all age groups and sexes except those aged 20 to 29. Notably, although absolute stroke mortality was higher among men in most years, women consistently exhibited higher proportional mortality. The joinpoint regression analysis further identified significant inflection points over the 20 years, highlighting specific intervals of either acceleration or deceleration in these trends.

Our findings align with previous research. For instance, De Moraes Bernal et al. (2020) [20] confirmed the decreasing trend in stroke mortality across various regions and age groups in Brazil. Similarly, Da Silva Paiva et al. (2021) [13] observed a reduction in proportional mortality among young adults nationwide from 1997 to 2012. More recently, [21] analyzed mortality trends from 2000 to 2018, reporting a consistent downward trend in both coronary heart disease and stroke mortality in Brazil. The decline in stroke mortality rates could be partly attributed to advancements in managing modifiable risk factors through outpatient care, as well as improvements in prehospital and inpatient care for acute stroke events [22].

Our study corroborates these trends, demonstrating that stroke proportional mortality trends in Espírito Santo are consistent with those observed in other Brazilian states, such as Piauí and Pernambuco, which have also experienced sustained declines from 2000 to 2018 [8]. On an international level, findings derived from the Global Burden of Disease Study substantiate a notable reduction in age-standardized mortality rates associated with stroke; however, they underscore that stroke continues to be a predominant cause of mortality, especially within low- and middle-income nations such as Brazil [22]. By directing attention towards Espírito Santo, a relatively smaller and less frequently examined state this investigation contributes regional perspectives that enhance the national and global comprehension of stroke mortality trends.

Implementing the National Stroke Care Project in Brazil likely contributed to the observed decline in proportional mortality from stroke across several states [6]. Specifically, in Espírito Santo, establishing a stroke unit may have significantly contributed to the decrease in proportional mortality among adults. Evidence shows that these units are well-known for substantially reducing mortality. Additionally, the Ministry of Health’s HIPERDIA system played a crucial role in managing patients with diabetes and hypertension within primary care. Hypertension is a well-established risk factor for stroke, and ample evidence supports this link [22]. An ecological study by Lopes et al. (2016) [23] documented a drop in hospital admissions for stroke, implying that implementing the HIPERDIA program may have contributed to the decline in stroke proportional mortality. However, it must be noted that evidence presented by Feigin et al. (2023) [24] has pointed to an increase in stroke incidence among young and middle-aged adults (ages > 55). Interestingly, our findings reveal that the age group 20–29 did not experience the same decline in mortality as other groups analyzed. Although the absolute numbers are low, it is important to highlight this aspect since it involves young adults, a population group that typically has lower mortality rates and should be a focus for prevention efforts.

The COVID-19 pandemic posed unique challenges to stroke diagnosis and treatment. During this period, the diagnosis of cerebrovascular events, such as strokes, decreased substantially compared to pre-pandemic levels [25,26]. Despite the relatively stable number of stroke-related deaths, there was a noticeable decline in proportional mortality among adults in Espírito Santo, with the lowest levels recorded in 2021. This decline may be attributed to the population adhering to social distancing recommendations, which led many to avoid hospitals and clinics, resulting in reduced demand for care for various conditions, including strokes [27].

However, the confinement and restrictions imposed during the COVID-19 pandemic likely had a dual impact on stroke outcomes. On the one hand, these measures might have prevented timely medical attention for individuals experiencing cerebrovascular events, exacerbating the severity of untreated strokes. On the other hand, the pandemic’s focus on COVID-19 may have disrupted the regular monitoring and reporting mechanisms for different diseases, including stroke, potentially leading to underreporting. Supporting this, Stein et al. (2021) [28] found that psychosocial stress, lockdown measures, and poor utilization of routine medical care during the pandemic worsened traditional stroke risk factors. Additionally, Sharma et al. (2021) [29] reported that a reduction in stroke-related emergency calls across 23 states and New York City was associated with an increase in excess stroke deaths.

Advanced age is associated with a higher incidence of strokes and an increased risk of more severe cases [30]. Our findings, which indicate that the highest proportional mortality occurs in individuals aged 70 and older, are consistent with existing evidence linking aging with an increased risk of stroke [31,32]. High mortality among older individuals due to stroke has also been observed in other Brazilian regions [33]. High public healthcare costs are driven by the long-term care requirements of stroke survivors’ chronic conditions [34].

Previous research conducted by Da Silva Paiva et al. (2019) [13] has indicated that proportional mortality from stroke was higher in women than in men during the period from 1997 to 2012, a trend that is also observed in the present study. Similarly, a systematic review conducted in the last decade reported that women had a higher percentage of stroke-related deaths compared to men (10.9% versus 6.6%). Another study by de Souza et al. (2024) [14], which analyzed stroke mortality in the metropolitan area of Greater Vitória from 2000 to 2021, also found that proportional mortality was higher among women. Data from the 2017 Global Burden of Disease (GBD) study further supported this, demonstrating that proportional mortality from cerebrovascular disease was higher in women, exceeding 30%, primarily due to population growth and aging. Our findings are consistent with those from the INSTRUCT (International Stroke Outcomes Study), where a meta-analysis of individual participant data from 13 population-based incidence studies showed that women had a higher crude mortality rate ratio compared to men at both one- and five-years post-stroke [35]. A possible explanation for this disparity is that women are more vulnerable to stroke due to the frequent presentation of atypical symptoms, which can make early diagnosis more challenging [36].

Despite the global and national decline in stroke mortality observed in this and other studies [22,37], stroke remains one of the leading causes of death and disability worldwide. Over the past two decades, Brazil and several other countries have implemented broad public health interventions aimed at prevention, early detection, and improved access to treatment [38,39,40]. These initiatives, such as smoking cessation programs, hypertension control, and promotion of healthy lifestyles, have contributed to important gains [41]. However, accurately assessing the burden of stroke still faces major challenges, particularly related to the quality and consistency of health data. Robust surveillance systems, standardized cause-of-death classifications, and improved data collection are essential to refine burden estimates [42,43].

This study uses data from the official Ministry of Health database, which provides comprehensive and representative mortality information for the state of Espírito Santo. The 22-year timeframe (2000–2021) allows for the identification of long-term trends in stroke mortality, offering valuable insights into the effectiveness of public health efforts. A notable limitation, however, is the potential underreporting of stroke cases and deaths during the COVID-19 pandemic, as individuals may have avoided seeking care due to infection fears. Moreover, because this is an observational study based on secondary data, it can reveal associations but cannot confirm causal relationships between interventions and mortality outcomes. Finally, while proportional mortality is a useful metric especially in settings with limited data it does not reflect the absolute risk of dying from stroke. Its interpretability can be compromised by factors such as underreporting and misclassification.

## 5. Conclusions

Between 2000 and 2021, proportional stroke mortality among adults declined significantly. This downward trend was observed in both sexes, with a reduction across all age groups except those aged 20 to 29. Throughout the study period, females consistently exhibited higher proportional mortality from stroke compared to males. These findings underscore the critical significance of ongoing public health initiatives aimed at stroke prevention and management, particularly within the younger adult demographic where no advancements have been evident. Considering the study’s dependence on secondary mortality data, augmenting the completeness of data—especially regarding variables such as sex—would significantly improve subsequent trend analyses. It is advisable that further investigations be conducted to examine the root causes of enduring disparities by sex and the stagnation of decline within younger populations.

## Figures and Tables

**Figure 1 epidemiologia-06-00028-f001:**
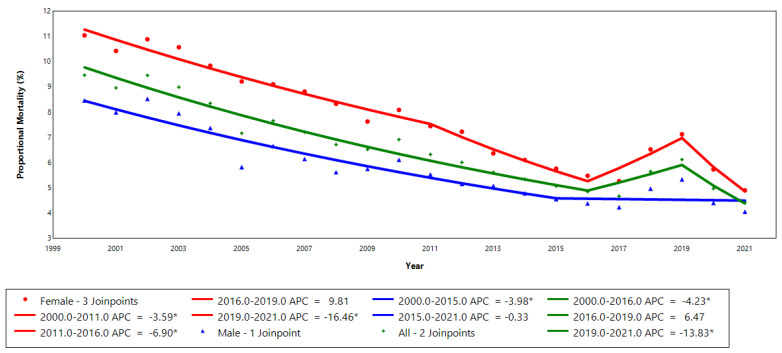
Annual percentage change in proportional mortality due to stroke in all adults and stratified by sex in Espírito Santo, 2000–2021. * *p* ≤ 0.05. APC: annual percentage change.

**Figure 2 epidemiologia-06-00028-f002:**
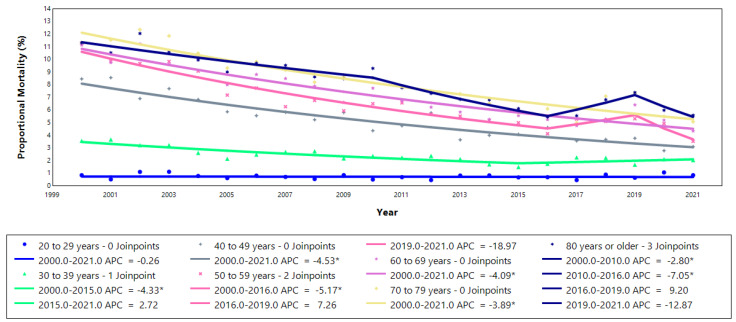
Annual percentage change in proportional mortality due to stroke in adults stratified by age group in Espírito Santo, 2000–2021. * *p* ≤ 0.05. APC: annual percentage change.

**Table 1 epidemiologia-06-00028-t001:** Number of deaths from stroke in adults in Espírito Santo by sex and age group, 2000–2021.

	All	Sex		Age Group						
		Male	Female	20–29 Years	30–39 Years	40–49 Years	50–59 Years	60–69 Years	70–79 Years	80 Years or +
	n	n	n	n	n	n	n	n	n	n
2000	1482	808	674	10	48	159	226	314	361	364
2001	1417	763	653	6	49	155	204	262	376	365
2002	1497	816	681	14	42	127	193	308	413	400
2003	1496	796	700	14	41	145	198	273	421	404
2004	1419	760	658	10	33	133	207	285	381	370
2005	1222	601	621	8	27	104	168	226	340	349
2006	1375	710	664	11	33	106	186	249	374	416
2007	1298	665	633	10	36	115	156	242	344	395
2008	1263	631	632	8	39	101	177	231	313	394
2009	1231	638	593	13	28	110	155	197	339	389
2010	1363	716	647	7	32	84	175	235	341	489
2011	1257	645	612	9	30	88	181	210	316	423
2012	1209	610	599	6	31	109	160	200	301	402
2013	1132	598	534	10	27	66	154	190	295	390
2014	1095	564	531	10	22	70	147	171	266	409
2015	1059	545	514	7	19	67	143	195	242	386
2016	1044	540	504	7	21	80	119	198	268	351
2017	1064	555	509	5	28	62	153	192	221	403
2018	1256	623	632	8	25	60	149	215	323	476
2019	1422	698	723	6	18	63	155	280	350	550
2020	1389	700	689	11	27	58	175	291	324	503
2021	1399	710	689	8	30	77	144	284	341	515

**Table 2 epidemiologia-06-00028-t002:** Proportional mortality due to stroke in adults in Espírito Santo, by sex and age group, 2000–2021 (values expressed as percentages).

	All	Sex		Age Group						
		Male	Female	20–29 Years	30–39 Years	40–49 Years	50–59 Years	60–69 Years	70–79 Years	80 Years or +
	PM	PM	PM	PM	PM	PM	PM	PM	PM	PM
2000	9.47	8.46	11.04	0.8	3.6	8.5	11.3	11.1	11.3	11.3
2001	8.96	8.00	10.42	0.5	3.7	8.6	9.9	9.8	11.5	10.5
2002	9.46	8.53	10.89	1.1	3.2	6.9	9.7	11.2	12.4	12.0
2003	8.99	7.95	10.57	1.1	3.2	7.7	9.8	9.6	11.9	10.6
2004	8.35	7.38	9.84	0.8	2.6	6.8	9.1	10.0	10.5	10.0
2005	7.17	5.83	9.22	0.6	2.1	5.9	7.2	8.0	9.3	9.0
2006	7.66	6.66	9.11	0.8	2.5	5.5	7.7	8.8	9.8	9.7
2007	7.22	6.15	8.82	0.7	2.7	5.8	6.3	8.5	9.2	9.5
2008	6.72	5.63	8.33	0.5	2.7	5.2	6.8	7.9	8.2	8.6
2009	6.53	5.76	7.63	0.8	2.2	5.8	5.9	6.6	8.6	8.5
2010	6.92	6.12	8.09	0.5	2.4	4.4	6.5	7.7	8.5	9.3
2011	6.33	5.54	7.45	0.7	2.2	4.7	6.6	6.7	7.9	7.8
2012	6.01	5.16	7.23	0.5	2.4	5.8	5.7	6.2	7.4	7.3
2013	5.62	5.09	6.37	0.8	2.1	3.6	5.5	5.8	7.3	6.8
2014	5.35	4.79	6.11	0.8	1.7	4.0	5.2	5.3	6.5	6.8
2015	5.07	4.56	5.77	0.7	1.5	4.1	5.0	5.6	5.8	6.1
2016	4.86	4.39	5.49	0.7	1.7	4.6	4.1	5.2	6.1	5.4
2017	4.68	4.24	5.27	0.5	2.2	3.6	5.3	4.8	5.0	5.5
2018	5.66	4.98	6.53	0.9	2.2	3.7	5.1	5.3	7.1	6.8
2019	6.13	5.34	7.13	0.6	1.7	3.8	5.3	6.4	7.3	7.4
2020	4.98	4.41	5.74	1.1	2.1	2.8	4.9	5.2	5.5	6.0
2021	4.44	4.07	4.91	0.8	2.0	3.1	3.5	4.3	5.1	5.6

**Table 3 epidemiologia-06-00028-t003:** Average annual percentage change in proportional mortality due to stroke in adults in Espírito Santo, 2000–2021.

	Average Annual Percentage Change (CI95%)	*p*–Value	Interpretation
All	−3.7 (−5.8; −1.7)	≤0.001	Decrease
Sex			
Male	−3.0 (−4.2; −1.7)	≤0.001	Decrease
Female	−3.9 (−5.6; −2.2)	≤0.001	Decrease
Age Group			
20 to 29 years	−0.3 (−2.2; 1.7)	0.780	Stability
30 to 39 years	−2.4 (−4.2; −0.5)	0.012	Decrease
40 to 49 years	−4.5 (−5.3; −3.8)	≤0.001	Decrease
50 to 59 years	−4.9 (−8.4; −1.4)	0.007	Decrease
60 to 69 years	−4.1 (−4.7; −3.4)	≤0.001	Decrease
70 to 79 years	−3.9 (−4.6; −3.2)	≤0.001	Decrease
80 years or +	−3.4 (−5.9; −0.9)	0.009	Decrease

## Data Availability

All data were retrieved from the Brazilian Ministry of Health’s database. For details regarding the data analysis, please contact the author at luiz.abreu@ufes.br.

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
