# Peer review of "Proportional Stroke Mortality in Espírito Santo, Brazil: A 20-Year Joinpoint Regression Study"

_epidemiologia, 2025, doi:10.3390/epidemiologia6020028_

Round 1
Reviewer 1 Report
Comments and Suggestions for Authors
This paper presents an insightful and well-executed ecological time-series analysis evaluating stroke-related proportional mortality trends in the state of Espírito Santo, Brazil, over a 20-year period. This is a clear research objective and strong public health relevance. The research is timely, methodologically sound, and relevant to public health policy in Brazil and globally. However, some aspects require improvement and enhancement:
- It is obvious that these figures and tables are not clear enough. The way they are inserted needs to be adjusted.
- The descriptions in some places are quite confusing, for examples:
Among males, there was a 4% monthly decline (p < 0.05) in proportional mortality 165 from 2000 to 2015, followed by a period of stability. Is this for every month? Or every year?
Although it represents only about 2% of the country's population Netto Djaló et al., (2024). Missing punctuation after “population.”
- The explanation of Joinpoint regression should be more precise and clear.
- In the discussion section, it is possible to briefly explore the potential underreporting and limitations of the proportion mortality rate as an epidemiological metric.
- Suggest professional English editing to enhance fluency and ensure grammatical precision throughout.
Again, this paper represents a valuable contribution to stroke epidemiology and public health surveillance in Brazil.
Comments on the Quality of English LanguageImprovement is needed. For example: The authors declare not conlict of interests.
After revision for language, statistical clarity, and others, it would be well-suited for publication.
Author Response
Reference: Manuscript ID: epidemiologia-3580545
Title: PROPORTIONAL STROKE MORTALITY IN ESPÍRITO SANTO, BRAZIL: A 20-YEAR
JOINPOINT REGRESSION STUDY
Dear Editor-in-Chief and Reviewers
We would like to thank the reviewers and the editor for their thorough evaluation of our manuscript and for the constructive comments and suggestions provided. These insights have been instrumental in improving the overall clarity and quality of our work.
We have addressed each point individually in the responses below. All changes made to the manuscript have been highlighted in yellow within the original file to facilitate review. We hope that the revised version meets your expectations and addresses all concerns satisfactorily.
Sincerely,
The authors
Revisor 1:
Open Review
(x) I would not like to sign my review report
( ) I would like to sign my review report
Quality of English Language
(x) The English could be improved to more clearly express the research.
( ) The English is fine and does not require any improvement.
|
Yes |
Can be improved |
Must be improved |
Not applicable |
|
|
Does the introduction provide sufficient background and include all relevant references? |
(x) |
( ) |
( ) |
( ) |
|
Is the research design appropriate? |
(x) |
( ) |
( ) |
( ) |
|
Are the methods adequately described? |
( ) |
(x) |
( ) |
( ) |
|
Are the results clearly presented? |
(x) |
( ) |
( ) |
( ) |
|
Are the conclusions supported by the results? |
(x) |
( ) |
( ) |
( ) |
|
Are all figures and tables clear and well-presented? |
( ) |
(x) |
( ) |
( ) |
Comments and Suggestions for Authors
This paper presents an insightful and well-executed ecological time-series analysis evaluating stroke-related proportional mortality trends in the state of Espírito Santo, Brazil, over a 20-year period. This is a clear research objective and strong public health relevance. The research is timely, methodologically sound, and relevant to public health policy in Brazil and globally. However, some aspects require improvement and enhancement:
Comment 1: It is obvious that these figures and tables are not clear enough. The way they are inserted needs to be adjusted.
Response: The figures have been revised and reformatted to improve clarity and understanding. To enhance the logical flow, each figure or table is now presented first, followed immediately by a detailed explanation of its content in the main text.
Comment 2: The descriptions in some places are quite confusing, for examples:
Among males, there was a 4% monthly decline (p < 0.05) in proportional mortality from 2000 to 2015, followed by a period of stability. Is this for every month? Or every year?
Although it represents only about 2% of the country's population Netto Djaló et al., (2024). Missing punctuation after “population.”
Response:
Comment 2.1: Among men, there was a statistically significant annual decline of -4.0% (p < 0.05) in proportional mortality from 2000 to 2015, followed by a period of stability.
Comment 2.2:
It was corrected.
Comment 3: The explanation of Joinpoint regression should be more precise and clear.
Response: Additional text was inserted in the Methods section to expand the explanation of the Joinpoint regression technique, including its purpose, structure, and relevance to trend analysis in epidemiological studies, as follows:
Joinpoint regression models were employed to identify significant changes in the time series data and to determine the trend within each segment of proportional mor-tality from 2000 to 2021(LACKLAND DT et al, 1999). This method is commonly used in epidemiological studies to analyze trends in disease incidence and mortality (Li & Du, 2020; Ghasemi et al., 2024). It identifies specific points in time—called join-points—where a statistically significant change in the trend occurs (Rea et al., 2017). The model fits a series of connected linear segments to the time series data, with each segment representing a distinct trend. This approach allows researchers to detect changes in the direction or rate of trends over time, such as periods of acceleration, deceleration, or reversal. The Jointpoint Regression Program selects the most ap-propriate model, using the Bayesian Information Criterion (BIC).
BIC helps choose the best model by favoring those that explain the data well without adding unnecessary joinpoints. Lower BIC values indicate better models.For each segment, 95% confidence intervals were calculated using the parametric method, and statistical significance was defined as a p-value ≤ 0.05.
Comment 4: In the discussion section, it is possible to briefly explore the potential underreporting and limitations of the proportion mortality rate as an epidemiological metric.
Response: The following text has been added at the end of Discussion section to address this point.
However, in conditions like stroke, accurately estimating disease burden requires improved data collection, standardized cause-of-death classification, and robust health information systems (Feigin et al., 2015; Cahuana-Hurtado et al., 2022).
While the Proportional Mortality Rate offers a simple and useful way to understand the relative importance of different causes of death—especially in settings with limited data — it does not reflect the absolute risk of dying from a specific cause. Its reliability is also affected by underreporting and misclassification of deaths. Therefore, PMR should be interpreted with caution and ideally supplemented by other epidemiological indicators when estimating the true burden of diseases like stroke. A key strength of this study is that it draws from the official Ministry of Health database, , which includes comprehensive and representative data from the state of Espírito Santo. The analysis spans over two decades (2000–2021), enabling the identification of long-term trends in stroke mortality and offering valuable insight into the effectiveness of public health policies. Despite these strengths, certain limitations must be acknowledged. For instance, during the COVID-19 pandemic, significant underreporting of stroke cases and deaths may have occurred, as individuals potentially avoided healthcare settings due to fear of infection. Further-more, as an observational study using secondary data, our analysis identifies associations but cannot establish causal relationships between public health interventions and mortality outcomes.
Comment 5: Suggest professional English editing to enhance fluency and ensure grammatical precision throughout.
Response: Thank you for the suggestion. The manuscript has been carefully revised to improve fluency and ensure grammatical accuracy throughout the text.
Again, this paper represents a valuable contribution to stroke epidemiology and public health surveillance in Brazil.
Comments on the Quality of English Language
Improvement is needed. For example: The authors declare not conlict of interests.
After revision for language, statistical clarity, and others, it would be well-suited for publication.
Reviewer 2 Report
Comments and Suggestions for Authors I believe this study presents important findings; however, several concerns need to be addressed. ⸻ 1. Introduction •Please clearly state the research hypothesis. ⸻ 2. Methods 2.1. Study Design and Population •Please clearly describe the inclusion and exclusion criteria for the study data. •Please specify how missing data were handled. •A flowchart of the participant selection process should be included, as this relates to the risk of selection bias. 2.2. Data Extraction •Please explicitly list which variables were extracted from the database. 2.3. Study Variables •You state: “Stroke-related death was defined according to the criteria of the American Heart Association and the American Stroke Association (Sacco et al., 2013). According to these criteria, stroke is defined as a neurological deficit resulting from an acute focal injury to the central nervous system of vascular origin and includes ischemic stroke, intracerebral hemorrhage, and subarachnoid hemorrhage.” •However, the pathophysiological mechanisms of ischemic stroke, intracerebral hemorrhage, and subarachnoid hemorrhage differ substantially. The current description does not sufficiently justify the appropriateness of analyzing them as a single, unified outcome. •It is necessary to conduct separate analyses by stroke subtype or perform sensitivity analyses with stroke subtype classification clearly stated. 2.4. Statistical Analysis •Please include a description of the sample size and how it was determined. ⸻ 3. Results •In the descriptions of Tables 1 and 2, the authors compare values across years. However, no statistical procedures appear to have been applied to test for significance; the analysis is limited to simple trends in numerical values. Please conduct appropriate statistical comparisons before drawing conclusions. •The same issue applies to comparisons between sexes. ⸻ Table 1: Please verify the data, as some subtotals do not match the overall totals (e.g., by sex in 2001). Table 2: Please specify the unit used for proportional mortality.Author Response
Revisor 2:
Open Review
(x) I would not like to sign my review report
( ) I would like to sign my review report
Quality of English Language
( ) The English could be improved to more clearly express the research.
(x) The English is fine and does not require any improvement.
|
Yes |
Can be improved |
Must be improved |
Not applicable |
|
|
Does the introduction provide sufficient background and include all relevant references? |
( ) |
(x) |
( ) |
( ) |
|
Is the research design appropriate? |
( ) |
(x) |
( ) |
( ) |
|
Are the methods adequately described? |
( ) |
( ) |
(x) |
( ) |
|
Are the results clearly presented? |
( ) |
( ) |
(x) |
( ) |
|
Are the conclusions supported by the results? |
( ) |
(x) |
( ) |
( ) |
|
Are all figures and tables clear and well-presented? |
( ) |
(x) |
( ) |
( ) |
Comments and Suggestions for Authors
I believe this study presents important findings; however, several concerns need to be addressed.
Comment 1.Introduction •Please clearly state the research hypothesis.
Response: Based on existing national and international evidence, we hypothesized that proportional mortality due to stroke would decrease over time, with variations across sex and age groups.
Comment 2. :
Comment 2.1 Study Design and Population. Please clearly describe the inclusion and exclusion criteria
Response: As this is a study based on secondary data from the DATASUS public health database, the inclusion and exclusion criteria are limited to the variables available in the system. The inclusion criteria consisted of all registered stroke-related deaths (ICD-10 codes I60–I64) among individuals aged 20 years and older, residing in the state of Espírito Santo, Brazil, between 2000 and 2021.
The exclusion criteria were records for individuals younger than 20 years or cases with missing geographic information that would prevent confirming residency in Espírito Santo.
Comment: Please specify how missing data were handled:
Response: The next text was included in the section Method ( Sub-section 2.5)
To address missing data, we applied complete case analysis, following recom-mendations by Stavseth et al. (2019), which consider this approach acceptable when the dataset exceeds 1,000 observations and the proportion of missing data is under 20%. Given the minimal proportion of missing cases, the application of this method is unlikely to have introduced bias or affected the validity of the results.
Comment: A flowchart of the participant selection process should be included, as this relates to the risk of selection bias
Response: We appreciate the reviewer’s comment. However, this is an ecological time-series study based entirely on secondary, publicly available mortality data from the DATASUS database. The dataset includes all stroke-related deaths (ICD-10: I60–I64) among individuals aged 20 years and older residing in Espírito Santo between 2000 and 2021, without any individual-level selection, recruitment, or sampling. As this analysis does not involve primary data collection, subject tracking, or case exclusions beyond standard filters applied at the database query level, a flowchart is not typically used in this type of design. We remain open to including a visual summary of the inclusion process if the editorial team considers it essential.
Comment: 2.2. Data Extraction •Please explicitly list which variables were extracted from the database.
Response: Thank you for your comment. As this is an ecological study based on secondary mortality data, the primary variable extracted was the underlying cause of death, specifically stroke-related mortality, identified using ICD-10 codes I60–I64. Additional variables extracted for stratification purposes included year of death, sex, age group, and place of residence (limited to Espírito Santo state). These variables were selected based on their availability in the DATASUS TabNet system and their relevance to the study objectives. We have clarified this in the revised version of the Methods section.
Comment: 2.3. Study Variables •You state: “Stroke-related death was defined according to the criteria of the American Heart Association and the American Stroke Association (Sacco et al., 2013). According to these criteria, stroke is defined as a neurological deficit resulting from an acute focal injury to the central nervous system of vascular origin and includes ischemic stroke, intracerebral hemorrhage, and subarachnoid hemorrhage.” •However, the pathophysiological mechanisms of ischemic stroke, intracerebral hemorrhage, and subarachnoid hemorrhage differ substantially. The current description does not sufficiently justify the appropriateness of analyzing them as a single, unified outcome. •It is necessary to conduct separate analyses by stroke subtype or perform sensitivity analyses with stroke subtype classification clearly stated.
Response:
Thank you for this valuable observation. We recognize that ischemic stroke, intracerebral hemorrhage, and subarachnoid hemorrhage have distinct pathophysiological mechanisms. However, in ecological time-series studies based on secondary mortality data, it is methodologically consistent and accepted to analyze stroke as a unified outcome using the standard ICD-10 coding block (I60–I64). This classification reflects how the Brazilian Ministry of Health aggregates and reports stroke mortality in the public health surveillance system. Therefore, our use of this aggregated definition is aligned with national epidemiological reporting standards and allows for comparability with other population-level studies using DATASUS data.
Hence we added the next text to the corresponding sub-section for transparency:
Although the pathophysiological mechanisms of these stroke subtypes differ, it is methodologically consistent and accepted in ecological time-series studies to analyze them as a unified outcome (DA SILVA PAIVA et al. 2019, DE SOUZA, O. F. et al.2024). This approach enables comparability with another population-level research using the same database. Therefore, stroke was analyzed as a single outcome variable in line with the study's ecological design and population-level objectives.
Comment 2.4. Statistical Analysis •Please include a description of the sample size and how it was determined.
Response:
As this is a population-based ecological time-series study using secondary data from the DATASUS system, no prior sample size calculation was performed. The study included all registered stroke-related deaths (ICD-10: I60–I64) among adults aged 20 years and older residing in Espírito Santo from 2000 to 2021. Thus, the analysis was based on a census of all eligible cases, not a selected sample. This information is included in the Methods section under “Study Design and Population” for clarity.
Comment 3 : Results •In the descriptions of Tables 1 and 2, the authors compare values across years. However, no statistical procedures appear to have been applied to test for significance; the analysis is limited to simple trends in numerical values. Please conduct appropriate statistical comparisons before drawing conclusions. •The same issue applies to comparisons between sexes.
Response: Tables 1 and 2 present descriptive data only, intended to illustrate the absolute values and variations across the years. For this reason, no statistical tests were applied to these tables, as they serve purely descriptive purposes.
The statistical analyses of temporal trends, including the corresponding hypothesis testing, are fully addressed in Table 3 and Figures 1 and 2, where Joinpoint regression was applied to identify significant changes over time.
Comment: Table 1: Please verify the data, as some subtotals do not match the overall totals (e.g., by sex in 2001).
Response: Thank you for your observation. We reviewed the data and confirmed that the discrepancy between sex-specific subtotals and the overall total in certain years is due to a small number of records missing sex information in the database.
To ensure transparency, we have added the following sentence at the beginning of the Results section:
“From 2000 to 2021, a total of 26,907 deaths from stroke occurred in Espírito Santo. For the variable sex, five records lacked information on the deceased’s sex, specifically in the years 2001, 2004, 2006, 2018, and 2019.”
Comment: Table 2, Please specify the unit used for proportional mortality.
Response: We would like to clarify that the unit of measure for proportional mortality is explicitly stated in the title of Table 2, which reads: "Proportional mortality due to stroke in adults in Espírito Santo, by sex and age group, 2000–2021 (values expressed as percentages)." We believe this provides clear information to the reader regarding how PM is represented in the table.
Reviewer 3 Report
Comments and Suggestions for Authors
The submitted study represents a quality contribution. It is well written and overall clear. The methodology is appropriate, the dataset is robust and the statistical analysis is valid. The manuscript addresses a relevant and timely public health topic. However, I have a couple of suggestions for improvement before publication:
- Discussion should be strengthened by critically comparing results with a wider range of regional and global studies.
- It is mentioned multiple times that the data is secondary and where it is acquired from. Revise to avoid redundancy.
- Formatting issues – reference list should be in MDPI style, references in the main text should be in square brackets. Tables should be in MDPI style tables provided in the template, not as an image or other format, as well as other inconsistencies. There are also a couple of typos throughout the manuscript.
- What about underreporting during COVID-19? Limitations section could be more robust
- What is novel in these findings compared to prior studies from other Brazilian states? Because you mention that your findings align with other studies.
- Conclusion needs to be revised. It needs to state what exactly is concluded from this study, are there any recommendations for further research, is there a need for better databases, clinical implications? Etc.
Author Response
Revisor 3:
Open Review
( ) I would not like to sign my review report
(x) I would like to sign my review report
Quality of English Language
( ) The English could be improved to more clearly express the research.
(x) The English is fine and does not require any improvement.
|
Yes |
Can be improved |
Must be improved |
Not applicable |
|
|
Does the introduction provide sufficient background and include all relevant references? |
(x) |
( ) |
( ) |
( ) |
|
Is the research design appropriate? |
(x) |
( ) |
( ) |
( ) |
|
Are the methods adequately described? |
(x) |
( ) |
( ) |
( ) |
|
Are the results clearly presented? |
(x) |
( ) |
( ) |
( ) |
|
Are the conclusions supported by the results? |
( ) |
( ) |
(x) |
( ) |
|
Are all figures and tables clear and well-presented? |
( ) |
( ) |
(x) |
( ) |
Comments and Suggestions for Authors
The submitted study represents a quality contribution. It is well written and overall clear. The methodology is appropriate, the dataset is robust and the statistical analysis is valid. The manuscript addresses a relevant and timely public health topic. However, I have a couple of suggestions for improvement before publication:
Comment 1: Discussion should be strengthened by critically comparing results with a wider range of regional and global studies.
Response : The next text is included in the Discussion Section:
Our findings align with previous research. For instance, De Moraes Bernal et al. (2020) confirmed the decreasing trend in stroke mortality across various regions and age groups in Brazil. Similarly, Da Silva Paiva et al. (2021) observed a reduction in proportional mortality among young adults nationwide from 1997 to 2012. More recently, Moreira et al. (2021) analyzed mortality trends from 2000 to 2018, reporting a consistent downward trend in both coronary heart disease and stroke mortality in Brazil. The decline in stroke mortality rates could be partly attributed to advancements in managing modifiable risk factors through outpatient care, as well as improvements in pre-hospital and inpatient care for acute stroke events (ARAUJO et al., 2024). Our study corroborates these trends, demonstrating that stroke proportional mortality trends in Espírito Santo are consistent with those observed in other Brazilian states, such as Piauí and Pernambuco, which have also experienced sustained declines from 2000 to 2018 (NETTO DJALÓ et al., 2024). On an international level, findings derived from the Global Burden of Disease Study substantiate a notable reduction in age-standardized mortality rates associated with stroke; however, they underscore that stroke continues to be a predominant cause of mortality, especially within low- and middle-income nations such as Brazil [3]. By directing attention towards Espírito Santo—a relatively smaller and less frequently examined state—this investigation contributes regional perspectives that enhance the national and global comprehension of stroke mortality trends.
Comment 2: It is mentioned multiple times that the data is secondary and where it is acquired from. Revise to avoid redundancy.
Response: We have revised the manuscript to eliminate redundant references to the data source and clarified this information in a single, appropriate section.
Comment:
Formatting issues – reference list should be in MDPI style, references in the main text should be in square brackets. Tables should be in MDPI style tables provided in the template, not as an image or other format, as well as other inconsistencies. There are also a couple of typos throughout the manuscript.
Response: The manuscript has been carefully edited to improve English grammar, clarity, and overall fluency throughout the text.
Comment 3: What about underreporting during COVID-19? Limitations section could be more robust
Response: The next text was included in the limitations
A notable limitation, however, is the potential underreporting of stroke cases and deaths during the COVID-19 pandemic, as individuals may have avoided seeking care due to infection fears. Moreover, because this is an observational study based on secondary data, it can reveal associations but cannot confirm causal relationships between interventions and mortality outcomes. Finally, while the Proportional Mortality Rate is a useful metric—especially in settings with limited data—it does not reflect the absolute risk of dying from stroke. Its interpretability can be compromised by factors such as underreporting and misclassification. Therefore, PMR should be used cautiously and complemented with other epidemiological indicators to more accurately estimate stroke burden.
Comment 4: What is novel in these findings compared to prior studies from other Brazilian states? Because you mention that your findings align with other studies.
Response: While our findings are consistent with national trends, the novelty of this study lies in its focus on Espírito Santo, a smaller state in the Southeast region that has been underrepresented in stroke mortality research. Most previous studies have focused on larger states or national-level data. By conducting a detailed, population-based time-series analysis over a 21-year period, our study provides important regional insights that can inform localized public health planning and stroke prevention strategies.
Comment 5: Conclusion needs to be revised. It needs to state what exactly is concluded from this study, are there any recommendations for further research, is there a need for better databases, clinical implications? Etc.
Response: The conclusion has been revised and expanded to clearly state the main findings, address the implications for public health and data quality, and provide recommendations for further research. See the updated version:
Between 2000 and 2021, proportional stroke mortality among adults declined significantly. This downward trend was observed in both sexes and across nearly all age groups, except for individuals aged 20 to 29. Throughout the study period, females consistently exhibited higher proportional mortality from stroke than males.
These findings support our hypothesis of a declining trend while revealing persistent disparities by sex and a lack of improvement in younger adults. These patterns point to the need for targeted public health interventions and deeper investigation into the factors driving these differences. Additionally, given the study’s reliance on secondary mortality data, enhancing the completeness and quality of key variables—such as sex—would strengthen future analyses

Round 2
Reviewer 2 Report
Comments and Suggestions for Authors
The manuscript states that “To address missing data, we applied complete case analysis, following recommendations by Stavseth et al. (2019), which consider this approach acceptable when the dataset exceeds 1,000 observations and the proportion of missing data is under 20%.” However, the proportion of missing data in this study is not reported anywhere in the text.
Please clarify the proportion of missing data in the dataset and justify the use of complete case analysis accordingly.
Author Response
Reference: Manuscript ID, Epidemiologia-3580545
Title: PROPORTIONAL STROKE MORTALITY IN ESPÍRITO SANTO, BRAZIL: A 20-YEAR
JOINPOINT REGRESSION STUDY
Second-Round Response
Dear Editor-in-Chief and Reviewers
We are grateful to all reviewers for their constructive comments, which have greatly improved the quality of our manuscript. At this stage, there are no pending comments from Reviewers 1 and 3. We now proceed to address the remaining minor comment from Reviewer 2, as outlined below.
We also confirm that the manuscript has been carefully revised for English language and grammar to ensure clarity and readability. Additionally, all changes made in response to the reviewer’s latest comments have been highlighted in yellow in the revised manuscript to facilitate review and ensure clarity.
Sincerely,
The authors
Reviewer 2 – Second Round
Comment s and Suggestions for Authors:
The manuscript states: “To address missing data, we applied complete case analysis, following recommendations [19], which consider this approach acceptable when the dataset exceeds 1,000 observations and the proportion of missing data is under 20%.” However, the proportion of missing data in this study is not reported anywhere in the text. Please clarify the proportion of missing data in the dataset and justify the use of complete case analysis accordingly.
Response:
We thank the reviewer for the comment. As noted in the Results section, only five cases were missing data for the variable sex. However, to improve clarity and transparency, we have now explicitly added the corresponding percentage (0.018%), as shown below:
Between 2000 and 2021, 26,907 stroke-related deaths were analyzed. The sex variable was missing in five records, corresponding to 0.018% of the dataset, specifically in the years 2001, 2004, 2006, 2018, and 2019. These omissions are considered negligible, with no documented evidence of systemic registration failures or clinical anomalies.
The justification for using complete case analysis—based on the minimal proportion of missing data and the total number of observations—was already included in the Methods section and remains as follows:
To address missing data, we applied complete case analysis, following established recommendations that support this method when datasets exceed 1,000 observations and the proportion of missing data is below 20% (Stavseth et al., 2019). Given the minimal number of missing cases (0.018%), the application of this method is unlikely to have introduced bias or affected the validity of the results.
